# Why Negative or Positive, If It makes Me Win? Dark Personality in Spanish Competitive Athletes

**DOI:** 10.3390/ijerph17103504

**Published:** 2020-05-17

**Authors:** Juan González-Hernández, Ricardo Cuevas-Campos, María Isabel Tovar-Gálvez, Lucía Melguizo-Rodríguez

**Affiliations:** 1Health Psychology/Behavioural Medicine, Research Group (CTS−0267), University of Granada, 18071 Granada, Spain; jgonzalez@ugr.es; 2Faculty of Education, University of Castilla La Mancha, 02071 Albacete, Spain; ricardo.cuevas@uclm.es; 3Department of Nursing, Faculty of Health Sciences, University of Granada, 18016 Granada, Spain; 4Biomedical Group (BIO277), Department of Nursing, Faculty of Health Sciences, University of Granada, 18016 Granada, Spain; luciamr@ugr.es

**Keywords:** narcissism, Machiavellianism, psychopathy, competitiveness, sport

## Abstract

Research on the dark triad traits (narcissism, Machiavellianism and psychopathy) is increasingly focusing on the functional or dysfunctional influences of personality traits on cognitive, behavioural and emotional responses. Thus, studies in sport contexts have shown that athletes who participate in competitive sports have higher scores in the dark triad than those who do not. The objectives of this cross-sectional study were to evaluate the linear and predictive relationships between dark traits and competitiveness (*p* < 0.01), as well as to identify any differences based on sports orientation (professionals vs. amateurs). Scales SD3 (dark personality) and C−10 (competitiveness) were applied to a sample of Spanish athletes (*n* = 806). The results show that competitiveness is strongly related to the traits of the dark personality triad. Narcissism is related to both the desire to win and the fear of losing, while Machiavellian tendencies are high when athletes feel like losers. Finally, psychopathic tendencies are related to feelings of inferiority and fear of failure. In conclusion, the results suggest that dark personality traits are related not only to the individuality of the athletes, but also to the self-perception of both their psychological response and the competitiveness of their sporting environment.

## 1. Introduction

Competitiveness can be described as a dynamical mental state in a context of rivalry, by which the athlete is pushed towards excellence due to a high component of social evaluation (or perception of social evaluation) [1]. In highly competitive circumstances, athletes’ performance and training are likely to be continuously evaluated.

In their research, Paulhus and Williams [2] defined the dark triad (DT) as a set of three personality traits so that “individuals with these traits share a tendency to be callous, selfish, and malevolent in their interpersonal dealings” (p. 100). The dark triad has been associated with personality traits that denote misbehavior, including Machiavellianism, which defines those individuals who feel the need to take revenge on the other [3], narcissism as a feature associated with excessive and exaggerated self-admiration with a hostile and aggressive tendency [4], and *psychopathy* as a personality feature characterized by an affective insensitivity, a lack of empathy and an absence of remorse, which is often linked to different types of criminality [5].

Sport Psychology recognizes the value of the dark traits in both research and practice [6]. Vaughan, R. et al. describe group differences in gender (men score higher than women), experience (athletes more experienced scoring higher than those with less expertise), and the type of sport (individual athletes score higher than team athletes across all factors) [7]. Other studies show that elite athletes have higher indicators of dark traits compared to amateur athletes who, in turn, scored higher compared to non-athletes [8].

Lochbaum et al. [9], in a meta-analysis of psychological correlations in competitive athletes, point out that those high in narcissism (e.g., excessive and exaggerated self-admiration) establish a connection to “undesirable behaviors”, with an absolute focus on achievement and the desire to win at any cost. Houston, Queen, Cruz, Vlahov and Gosnell [10] describe that athletes with high Machiavellian scores are associated with considerable levels of competitiveness and hyper-competitiveness, being the most important characteristic for improving confidence in one’s own abilities and psychological resource. Other studies describe the cost of performance-approach goals in terms of the use of avoidance strategies (i.e., cheating and reluctance to cooperate with peers) [11,12].

The study of dark triad traits (narcissism, Machiavellianism and psychopathy) (DT) [2] highlights the functional [13] and dysfunctional [14] value of these traits in the athlete and on those which have competitive versus non-competitive features on their own performance. This allows us to understand how personality traits (regardless of whether they are considered good or dark), relate to individual or social perceptions in sports environments for a balanced psychological adjustment [6], adequate performance [15], and a competitive experience or sport improvements for athletes [16].

For sustained physical activity, narcissism has been related with mental toughness [7], as well as pride and dominance [17]. Houston et al. [10], established a relationship between narcissism and commitment to achieving the goals themselves, and associated Machiavellian tendencies with an underestimation of the possible costs and “side effects”. In contrast, Nicholls, Madigan, Backhouse and Levy [18] showed dark traits, mainly Machiavellianism and psychopathy, correlated positively with attitudes towards doping.

From a psychosocial perspective, athletes’ competitiveness has been conceptualized as a potential resource in sports performance which influences the appearance of functional behaviour [19]. Martens [20] defined it *“as a willingness to strive for a standard of excellence when making comparisons in the presence of external evaluators. It constitutes a conduct of achievement in a competitive context, where social evaluation is the key component”* [21] (p. 168). From this perspective, athletes perceive and self-regulate all sports demands in interaction with their environments, developing intricate and complex connections between each component of this “ecological-like” system (sport competitive context), where personal goals, expectations and constraints (even business ones in some cases) are in permanent conflict to achieve success or avoid failure [22].

At the same time, sport is a vehicle in which it is possible to reflect competence vs. incompetence in front of others [23], or to generate beliefs and emotional experiences that, in fact, produce dysfunctional responses and have an impact on the individual´s performance [24]. For example, fear of failure, mainly associated with negative self-assessments of one´s own capacities [25], or vulnerable narcissism [26], has been considered, from the neurocognitive point of view, as a stable tendency to anticipate shame and humiliation after failure. Fears are translated into emotions when a person undergoes stimulus subjectively interpreted as dangerous, unknown, or uncontrollable [27]. This effect is mainly due to an overgeneralization of consequences [28]. Otherwise, the fear signal becomes an important resource to respond effectively (preservation behaviours). In sports situations, a proper interpretation of pressure implies an “preparing and activating alert”, focused on the most immediate action (occupying the mind), and based on beliefs of necessity [29] as well as on personal possibility [30].

Focusing on a cross-sectional, correlational and predictive design, this study aims to highlight the differences and relationships between dark triad traits, competitiveness, feelings of satisfaction (or dissatisfaction) with sports results and performance self-evaluations (feelings of winner and loser) in a sample of Spanish athletes. The analysis will show greater differences in favor of professional athletes vs. amateur athletes in terms of dark traits and competitiveness, while we expect the dark personality features to reveal positive relationships with competitiveness factors (motivation to succeed and motivation to avoid failure) and positive feelings (to be a winner and satisfaction with sport results). In contrast, dark traits will also turn out to be negative feelings linked to negative perceptions of loss.

## 2. Materials and Methods

### 2.1. Participants

The study sample consisted of 806 (483 males; 323 females) professional and non-professional (amateurs) athletes (age  =  25.1 years, SD  =  4.6, range  =  19 to 33 years) who competed in different sports (Table 1). Samples were recruited from clubs, federations and competitions events (i.e., popular races or tournaments). The average athletic experience in the practiced sports was 9.36 years. (SD  =  5.8). The average training time per week was 10.28 hr. (SD  =  2.3).

### 2.2. Instruments

Socio-demographic data. Information about age, sport practiced, time spent practicing sports and training time per week was collected, as well as questions about whether participants were satisfied or not with their sports results and whether they perceived themselves as winners *(“How much of a winner do you feel in your sports practice?”*) or losers *(“How much of a loser do you feel in your sports practice?”*) in their overall sport life [Likert scale from 1(“*no winner or loser at all”*) to10 (“*a lot winner or loser”*)].

Dark Triad Personality. The Short Dark Triad scale (SD3) [31] was used. This is a 27-item questionnaire that measures Machiavellianism (e.g., *“Make sure your plans benefit you, not others”*), narcissism (e.g., *“Many group activities tend to be dull without me”*), and psychopathy *(“People often say I’m out of control”*). All questions were answered on a 5-point Likert-type scale, which was anchored at 1 = *“not at all”* and 5 = *“extremely”*. The traits´s reliability showed significant internal consistency.

Competitiveness. The competitiveness Scale−10 (C−10) [21] was used. This is a self-report questionnaire with 10 questions about the motivation associated with sports competitiveness; it is composed of two first-order factors: motivation for success (MS) (e.g., *“I wish to be the best every time I compete”*) and motivation to avoid failure (MAF) (e.g., *“I’m worried about what others might think of my performance”*). The reliability factors were alpha Cronbach scores adequate both MS (0.84) and MFA (0.92).

### 2.3. Procedure

Following the same protocol, the questionnaires were administered randomly to the athletes in their own sport contexts (i.e., locker rooms, playing courts, facilities). Neither sport centres nor athletes were explicitly called up, but only those present on the training days agreed upon with the heads of the clubs or sports federations were administered in groups. Each questionnaire was answered individually and voluntarily with pencil and paper in the presence of at least one researcher to address any questions or concerns that might arise. At the same time, an informed consent form was attached, in which the athletes voluntarily agreed to participate in the study, and which described the measures taken to maintain the confidentiality and anonymity of the responses. All the measure protocol was in accordance with the ethical guidelines of the American Psychological Association and following the tenets of the Declaration of Helsinki of 1975, as revised in 2013 (Project identification code: Research Ethics Committee IEC/85/15/75 of 21 June 2019).

### 2.4. Analysis

Using the statistical program SPSS 23.0 (IBM Inc., Armonk, NY, USA) an analysis of distribution, description, normality (K-S) of the sample and internal consistency (Cronbach’s alpha) was performed in order to show the reliability of the questionnaires and the suitability of the use of parametric tests. To address those data that were not shown to be within the normal distribution, all estimates that follow are based on 5000 bootstrap samples with bias-corrected confidence intervals (95.00). In addition, the Box and Wilks’ lambda test was used for discriminant analysis to classify the variables in two categories (professional vs. amateur) (i.e., to examine whether there are significant differences among the groups, in terms of the predictor variables, evaluating, as well, the accuracy of the classification). Pearson’s bidirectional analysis and multiple regression analysis (stepwise) were also proposed for the statistical relationship (linear and predictive, respectively) between the study variables. The significance levels of this study were set at *p* < 0.05.

## 3. Results

### 3.1. Descriptive Sample Analysis

The most important sports in Spain were represented (Table 1), with footballers (soccer and futsal players) being the most numerous subsamples in both professional and amateur sport. The highest average age for professional athletes was in athletics, while for amateurs it was futsal players.

### 3.2. Discriminant Analysis

The results of Wilks Lambda [0.89; F_(1794)_ = 42.36 (<0.01); χ^2^: 38.63 (<0.01)] and Box [text 78.03; F_(1794)_ = 86.24 (<0.01)] tests, reflected that both variance and covariance are different. Canonical correlation was 0.41, showing that the eight psychological attributes explain 43% of the variance in the dependent variable, indicating the convenience of differentiating and categorizing between professional vs. amateur athletes. Table 2 showed that professional athletes significantly discriminated against in narcissism (<0.01), Machiavellianism (<0.01), psychopathy (<0.01), MAF (<0.01) and self-satisfaction with sport results (<0.01), while amateur athletes discriminated significantly against feelings of being a loser (<0.01).

### 3.3. Analysis Correlation

Pearson´s test analysis showed strong linear relationships between dark personality traits, self-assessments and competitiveness dimensions, showing interesting positive and negative links (Table 3). Athletes with the highest scores for narcissism (i.e., grandiosity), showed positive links with both MS, MAF, sport satisfaction and the feeling of being a winner, while negative relationships were established with feelings of being a loser. The Machiavellian trait (i.e., social manipulation) showed positive links with MS, MAF and feelings of being a loser, while negative links were with feelings of being a winner. Linear relationships also showed that, as the psychopathy trait (i.e., low empathy) was greater, MS, MAF and feelings of being a loser increased, while satisfaction with sports results decreased. While MS was associated with high satisfaction with sports results and feelings of being a winner, the opposite was observed for those athletes who showed greater efforts to avoid failure (lower satisfaction with their sports results and greater feelings of being a loser).

### 3.4. Multiple Regression Analysis

A stepwise regression analysis was proposed (Table 4), introducing covariates with the following sequence: competitiveness, general satisfaction and feelings about sport sensations. The results described that narcissism (43% of variance) [F_(1802)_ = 7.42; *p* < 0.02] was predicted under the increase of scores of MS and MAF, feelings of being a winner, competitiveness, satisfaction with sport results, and lower feelings of being a loser scores. Machiavellianism (27% of variance) [F_(1802)_ = 9.24; *p* < 0.01], was predicted, under the increase of both MS, MAF and feelings of being a loser, and the decrease of feelings of being a winner. Finally, psychopathy was predicted (18% of variance) [F_(1802)_ = 6.04; *p* < 0.05] under the increase in competitiveness, efforts to avoid failure and internalised criticism regarding competitive loss.

## 4. Discussion

The present study sought to examine whether competitiveness and satisfaction feelings and self-evaluation performance scores were significant predictors of dark features of personality (narcissism, Machiavellianism and psychopathy) in a sample of Spanish professional and amateur athletes. The results partially supported the hypothesis proposed about the existence of associations between most of predictive variables of narcissism, while Machiavellianism and psychopathy were shown to be related to the dimensions of competitiveness and the feeling of winning (Machiavellianism) and losing (psychopathy). In addition, the category of professional athletes was significant integrating narcissism, Machiavellianism, psychopathy, MFA and satisfaction with sports results, while the category of amateur athletes showed only differences in negative perceptions of loss.

### 4.1. Dark traits and Negative Relationships. Tributes for the Desire to Win

Research on dark personality traits in athletes [7] follows the same conductive line as current research in the general population [13,32] regarding personal and social self-representations (i.e., the importance of being superior or better than others). The results of the present study reinforce the differences between professionals and amateur athletes, adding competitiveness and performance self-perceptions. Furthermore, the results partially confirm the hypothesis initially raised about the dysfunctional relationship with the self-assesment of internalised criticism regarding competitive loss (while Machiavellian trait was negatively related to feelings of being a winner, Machiavellian and psychopathy traits showed positive relationships with feelings to being a loser and MFA).

The narcissistic trait has shown strong relationships with competitiveness factors (MS and MFA), satisfaction with sports results, and feelings of both being winners (positively) and losers (negatively). In this sense, studies have indicated a greater perception of social anxiety [33], lack of moral commitment [14] with high scores of narcissism in athletes.

The Machiavellian trait has been significantly linked to competitiveness factors (MS and MFA), and feelings of being losers (positively), and to feelings of being winners (negatively). In this sense, exposure to stressful or unstable environments during early sports experiences has often been marked as the path for expressions of dark traits expressions [34] or doping attitudes in athletes with significant Machiavellian scores [18].

Finally, psychopathic trait have been significantly and positively related to MS, MFA, and feeling of being a loser, while significantly negative relationships have been established with the satisfaction of sport results. Studies in high-performance athletes, have poor empathy [35,36] or lack emotional expression [37] when athletes are too focused to achieve their results. The difficulty in controlling the emotional impulses exaggerates the agony of the athletes for not being able to fulfil the established competitive commitments, moving him away of him of the satisfaction for fighting for them as well as the social relationship with the surrounding environment (i.e., solitude, conflicts) [38].

### 4.2. Dark Traits and Positive Relationships. Internal Net of Security for the Desire to Win

Studies have reported that people who participate in competitive sports have higher scores for the dark triad than amateurs [7,39], highlighting the emergence of these traits with greater intensity in high-performance athletes. In the present study, professional athletes indicated that they ranked higher and significantly in both narcissism, Machiavellianism and psychopathy, and in competitiveness (MS and MFA). Therefore, the results confirm the hypotheses of both a greater identification of dark traits in professional athletes and the functional relationship of dark traits to MS (positive competitiveness), satisfaction with sports results and feelings of being a winner.

Johnson, Panagioti, Bass, Ramsey, and Harrison [40], point out the relevance about the existence of adequate indicators of narcissism, like a protective factor of exercise addiction, when self-esteem is low and this allows us to assume an interesting link in the construction of self-esteem. Other authors consider the influence of a moderate or functional narcissism in relation to motivation [41] or mental toughness [7]. Machiavellianism offers a stronger relationship with the thought structure that the athletes should create (i.e., their beliefs of perfectionism) like their mental routines, or their relationships with rivals, partners or power figures [42]. It is also related to a tendency to dominate social relationships as well as with a trend to control or to need security, and these, in various domains of the environment. In this sense, high level of Machiavellianism is associated with high level of leadership [12], greater indicators of self-control, low social concerns and achievements [10].

Recent research on the positive characterization of personal traits in sport contexts [43,44,45], recommend exploring the dark personality features, as a source of an explanatory gap on psychological behaviour in sports, and as a complement of the currents of positive psychology. The constant exposure to a context of achievements makes the sports context lead athletes to translate their achievement into success, but also to make a mental effort to demonstrate their competence [46], ability [18], self-control [47], or their level of motor efficiency [48], associated with its non-achievement, when they perceived the failure.

Some limitations of the present study need to be addressed. The athletes within this sample were elite and non-elite, which probably implies differences in lifestyle, such as the fact that athletes with professional contracts (i.e., football or basket players) have different living arrangements compared to athletes with scholarships (i.e., equestrian or taekwondo). The use of three of the variables as individual and subjective elements may be an important limitation, but we consider that their inclusion was explicit and direct so as not to saturate athletes with the administration of measures. Differences in lifestyle and compensation may also influence the living arrangements, family and other commitments. No data on nearby competitions or classification systems (i.e., rankings) have been documented. This could be considered as a limitation because these elements are important to athlete’s interpretations of their feelings, thoughts or behaviours in upcoming competitions.

On the other hand, this research lays the basis for future research in the field of personality features, to provide a useful scientific information to understand many of psychological responses of athletes in competition situations. If research on personality in sport is conducted in this way, we could help to explain and indicate a large number of negative situations, while working at the same time on to describe risk factors, proposing psychological plans full of coherence and adjusted to the individual features of athletes (i.e., gender, age) and their contexts (i.e., type of sport, individual vs. team sports).

This study offers several indications on how to view the practice of sport, and, at the same time, how to understand dark features. Furthermore, it is the first study carried out on a sample of Spanish athletes. Although the connection of dark features with mental illness is undeniable, in sports contexts, research shows ambivalent influences towards positive (e.g., success motivation) or negative response (e.g., fear of failure). In any case, and similarly to goal-oriented research [49], this line of research can offer a wealth of information about the relevance of the search for success (or being competitive) and balance in mental health in athletes. For this reason, we consider the importance of defining the fine line between competitiveness, suffering and illness.

## 5. Conclusions

The present study confirms how personality traits (regardless of whether they are considered good or dark) related to the competitiveness of a sample of high-performance Spanish athletes, as well as with their self-perceptions and sports environments. This study highlights both the positive and negative connections of being competitive (an aspect that all athletes and their coaches seek to improve).

For these reasons, according to our point of view, it is worth going deeper into the nature of the construction of narcissism, Machiavellian or psychopathic thoughts and their relationship with the variables of competitiveness, would provide powerful factors to athletes about their mental resources for sports performance. However, the presence of dark traits can lead to the emergence of dysfunctional mental health (e.g., addiction), as the literature on dark traits has shown with other type of populations as well (e.g., those rating very high in these traits are likely to experience interpersonal problems with teammates or coaches etc. The authors need to be more sophisticated in their conclusion, and integrate what is known about the maladaptive aspects of narcissism, Machiavellianism and psychopathic traits).

Sport Psychology should continue to address the many and complex issues of relations between personality and sporting activity. In this way, the benefits would become useful for athletes in their performance, personal career, life planning, psychological adjustment, self-management (such as stress/time management) and improvement in interpersonal skills. Furthermore, in order to investigate in detail the mental functioning of athletes and effectively assess their personality traits within competitive reality, sports psychologists should focus on understanding competitiveness within a very high achievement environment, where it is necessary to regulate competitive traits and develop a deep understanding of sports relationships, situations and experiences.

## Figures and Tables

**Table 1 ijerph-17-03504-t001:** Sample distribution for sport practiced.

Sport	General Sample (%)	Age M(SD)	Professional Sample (%)	Age M(SD)	Amateur Sample (%)	Age M(SD)
Soccer	168(20.84)	24.6(3.5)	74(44.05)	21.8(4.3)	94(55.95)	26.6(2.7)
Rugby	46(5.71)	27.2(6.1)	21(45.65)	25.8(3.4)	25(54.35)	29.3(5.4)
Tennis	28(3.47)	22.4(3.6)	10(35.71)	23.3(5.1)	18(64.29)	20.7(5.8)
Volleyball	34(4.22)	23.5(6.4)	16(47.06)	22.9(2.9)	18(52.94)	26.4(3.4)
Basketball	69(8.56)	27.6(5.8)	32(46.38)	24.2(3.6)	37(53.62)	29.6(5.8)
Athletics	85(10.55)	28.4(3.5)	26(30.59)	26.3(7.2)	59(69.41)	30.1(4.2)
Swimming	74(9.18)	25.7(4.1)	25(33.78)	22.8(3.4)	49(66.21)	28.6(3.5)
Equestrian	27(3.35)	22.1(6.3)	7(25.93)	18.9(2.3)	20(74.07)	25.2(7.2)
Handball	96(11.91)	22.8(4.5)	61(63.54)	21.4(4.4)	35(36.46)	21.8(4.5)
Futsal	132(16.38)	28.4(7.1)	48(36.36)	26.2(5.2)	84(63.64)	30.4(7.1)
Taekwondo	47(5.83)	24.9(2.4)	20(42.55)	21.8(2.7)	27(57.45)	24.9(2.4)
	*n* = 806		*n* = 340		*n* = 466	

**Table 2 ijerph-17-03504-t002:** Discriminant analysis between groups (professional vs. amateurs) in dark personality competitiveness and sporting feelings patterns.

*n* = 806	Sum of Square	*d*	Professionals M(SD)	Amateurs M(SD)	(λ)	F	SC *p*-Value
Narcissism	4083.21	0.30	26.72(4.08)	25.04(4.83)	0.94	24.14	0.83 *
Machiavellianism	3874.76	0.26	28.56(6.14)	26.01(7.36)	0.93	11.43	0.72 **
Psychopathy	2723.61	0.34	20.17(5.09)	18.06(4.43)	0.89	4.02	0.63 **
MS	196.15	0.27	3.23(1.02)	3.04(0.46)			0.78 **
MAF	190.42	0.33	3.37(0.58)	2.84(1.19)	0.96	16.94	0.84 **
SSR	90.25	0.35	7.46(1.83)	5.34(3.01)	0.98	27.02	0.75 *
Fwinner	98.76	0.32	6.78(1.91)	6.23(2.48)			0.64 *
Floser	95.69	0.32	4.15(1.16)	5.60(1.04)	0.97	21.32	−0.69 **

* *p* < 0.05; ** *p* < 0.01; (λ): Wilks Lambda; SC < *p*-value = Standarized canonical coeficient; d: effect size. MS: motivation to succeed; MAF: motivation to avoid failure; SSR: satisfaction with sporting results; Fwinner: feeling like a winner; Floser: feeling like a loser.

**Table 3 ijerph-17-03504-t003:** Pearson´s correlations between dark personality traits, competitiveness and personal feelings about himself.

	K–S ^1^	1	2	3	4	5	6	7	8
Narcissism	0.22	(0.84) ^2^	0.56 **	0.62 **	0.73 **	0.62 **	0.76 **	0.78 **	−0.56 **
Machiavellianism	0.19		(0.79) ^2^	0.67 **	0.48 *	0.53 *	−0.56	−0.45 **	0.49 **
Psychopathy	0.24			(0.84) ^2^	0.43 *	0.47 *	−0.36 *	0.08	0.43 **
MS	0.20				(0.92) ^2^	−0.68 **	0.43 **	0.61 **	−0.48 *
MAF	0.21					(0.86) ^1^	−0.28 **	−0.55 *	0.67 **
SSR	0.20						-	0.69 **	−0.67 **
Fwinner	0.24							-	−0.63 **
Floser	23								-

^1^ K–S: Kolgomorov–Smirnov (<0.05). ^2^ Cronbach´s alpha. Bias-corrected and accelerated 95% confidence intervals. * *p* < 0.05; ** *p* < 0.01. MS: motivation to succeed; MAF: motivation to avoid failure; SSR: satisfaction with sporting results; Fwinner: feeling like a winner; Floser: Feeling like a loser.

**Table 4 ijerph-17-03504-t004:** Predictive analysis of dark personality traits in athletes, according their competitiveness.

*n* = 806		Coefficients			
		VI	β	*t*	*p*
Narcissism R^2^ = 0.43; (0.00)	stepwise 1	MS	0.52	4.78	**
		MAF	0.39	4.17	*
		Competitiveness	0.64	6.02	**
	stepwise 2	SSR	0.61	5.43	**
	stepwise 3	Fwinner	0.58	6.03	**
		Flooser	−0.51	5.16	*
Machiavellianism R^2^ = 0.27; (0.00)	stepwise 1	MS	0.38	4.04	*
		MAF	0.42	4.27	**
		Competitiveness	0.58	2.08	**
	stepwise 2	Fwinner	−0.31	3.14	**
		Flooser	0.42	4.63	**
Psychopathy R^2^ = 0.18; (0.00)	stepwise 1	MAF	0.42	2.86	**
		Competitiveness	0.52	4.05	**
	stepwise 2	Flooser	0.53	3.37	**

* *p* < 0.05; ** *p* < 0.01. Bias-corrected and accelerated 95% confidence intervals. MS: Motivation to succeed; MAF: Motivation to avoid failure; SSR: Satisfaction with Sporting Results; Fwinner: Feeling like a winner; Flooser: Feeling like a loser.

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
