# Peer review of "Why Negative or Positive, If It makes Me Win? Dark Personality in Spanish Competitive Athletes"

_ijerph, 2020, doi:10.3390/ijerph17103504_

Round 1

Reviewer 1 Report

This version of the paper is vastly improved form the original submission. The authors have clearly conducted a major revision and rewrite. However, while improved, there are still major concerns with the paper, and these are outlined below. One of my major concerns is about the specific new contribution this study makes to the field, and why the results are important? 

Introduction

Suggest removal of the inverted question mark in the article title given the article is in English text (as opposed to Spanish).  

Per DSM-5, “Psychopathy” is technically not a personality disorder. The closest disorder is antisocial personality disorder.

“…those with a high ego…” It is unclear what the authors mean by “high ego”.

“Most of the researches related…” should read “research”.

The content from Line 106 needs to be rewritten – if this is the hypothesis, then can it be more clearly stated as such? Eg., This study predicted that… The wording in this section in through to line 111 is unclear.

“…higher differences…” Do the authors mean larger differences?

“…negative feelings linked to being loser.” Suggest rewording here, and throughout the paper. Would terms like “negative perceptions of loss” or “internalised criticism regarding competitive loss” be more accurate?

I think there is a tendency to overuse quotations in the introduction section. Can some of these be reworded and removed? In general, quotes should be used sparingly and only when absolutely necessary.

Method

What was the difference between elite and non-elite? What definitions were used for these?

Table 1 – this should be moved to results section, and there should be some description of the sample.

Line 126 – inconsistency in italicisation of sample scale items.

The SD3 internal consistency values (alpha) should be reported rather than just stating there was internal consistency.

Section 2.3 – Which institution provided ethics approval, and what was the approval number? Please list.

Section 2.3 – how were sports identified to participate?

Section 2.3 – were data complete prior to, during, or after sport participation? This has implication as to whether these factors may have biased responses E.g., different patterns of response for athletes who had won / lost?

Line 121 – More information is needed for the items “Satisfaction with Sporting Results”, F loser and F winner in order to properly interpret what these mean in the results. What is the reference period for these; today, past week, past year?

Line 146 – “Participation was voluntary, responsible and honest.” The term responsible is not appropriate here. The researchers cannot determine is participation was “honest” without the inclusion of a lie scale. This should be deleted.

Results

There is no description offered for the sample re: socio-demographic characteristics.

I’m unclear about the analysis that is reported in Table 2. Is this discriminant function analysis or a MANOVA? If it’s a MANOVA, suggest changing the terminology from “Discriminant analysis” to “Between groups analysis” or similar. The term Discriminant analysis suggestion discriminant function analysis, and it’s not clear if that has been conducted as discriminant function analysis is about predicting to group membership. Table 2 looks like a MANOVA.

If a MANOVA was conducted, what is the rationale for including the non-dark triad variables (eg., MS: Motivation to success; MAF: Motivation to avoid failure; SSR: Satisfaction with Sporting Results; F winner: Wining Feeling winner; F looser: Loser Feelings)

Suggest including a measure of effect size like Cohen’s d to Table 2, to describe the magnitude to effects.

P-values cannot be “p<.00” even if SPSS reports this - this should be “p<.01” or the actual p-value written which SPSS provides if the cell is double-clicked in the output.

There is a typo in the table note “Winnig”

Table 3 – what are the values in brackets? Are they alphas? Needs a table note.

Table 3 – are the KS values the p-value or the KS test statistic? It is not clear if the scales normally distributed, and if not, what adjustments have been made for non-normality.  

Line 171 – “Athletes with the highest scores for narcissism (i.e., grandiosity), showed positive links with both…” This statement suggests that analyses were undertaken by high / low group, but this doesn’t seem to be the case.

Section 3.3 – this needs a proofread for grammar and restructure.

Section 3.2 and 3.3 – the authors establish amateur / professional differences in study outcomes in Table 2, but then the analyses data presented in Tables 3 and 4 are conducted without any consideration of these differences. Why not present separate correlations in Table 3 and separate regressions in Table 4 (or test for group membership as a predictor)?

Were there any predictive effects by sport type (e.g., individual versus team)? The introduction mentions that sport type can impact response for dark triad variables, but this is not examined in the present analyses, despite the sample being large enough to do so.

Discussion

The specific new contribution this study makes to the field is not entirely clear. Can the auhtors be specific about the new knowledge that is provided here, and why it is important? From the introduction, it seems like much of the information reported in this paper is already known to the field. I think the authors need to be clearer on what the new contribution is here. 

This may need to be re-written pending changes to the analysis.

Three of the variables were single items – this is major limitation as it is unclear how valid and reliable these measures are.

Line 217 – “Maquiavellian” is misspelt.

Line 282 – “The present study confirms how personality traits (regardless of whether they are considered good or dark) related to the individuality of a sample of high-performance Spanish athletes,” This is unclear; what do the authors mean by “individuality”.

Line 285 – the implications for sport psychology are vague; can the authors offer some specific suggestions for what these findings mean for the practice of sport psychology?

Reviewer 2 Report

I commend the authors for addressing the original concerns regarding the manuscript. The authors have provided the extra detail required and have produce an insightful manuscript. I still believe the conclusion could be strengthened further and encourage the authors to really identify how this information can be applied in context.

Reviewer 3 Report

IJERPH-793294 presents results from a sample of competitive athletes. While some parts of this manuscript were interesting, other areas could be improved. I hope the authors consider my feedback for enhancing their manuscript.

  • Please remove all comments in the right column.
  • Pleaser revise reference formatting to meet journal criteria.
  • Line 82: “Most of the researches related…” Please correct the manuscript for minor grammatical mistakes.
  • Lines 103-111: Referring to the analyses here takes away from the prospective nature of the manuscript.
  • The Introduction section is a little long and distracting to the reader. Can you please reduce text in this section so that it is more to the point?
  • Section 2.1: This section is more so appropriate for the Results. The participants section is more so for IRB approval, recruitment, experimental design, inclusion criteria, etc.
  • Line 159: What exactly is “(<.00)”? If this is a p-value, just round to <0.01.
  • Results (Table 4): While I understand the use of stepwise regression here, it remains unclear what covariates were included as a result of the stepwise procedure. More clarity is needed.
  • The R-squared values in Table 4 are still lower. Should the authors instead complete a pre-specified approach to their models in effort to improve variance explained?
  • Please insert a limitations paragraph in the Discussion.
  • The abstract needs some specific statistics to support the results.
  • Please make any changes to the abstract that align with those in the text.

Round 2

Reviewer 1 Report

While the authors have attended to many of the comments raised in the second review, many points have been insufficiently addressed or totally ignored. This makes me think that the present revision is relatively 'lazy', and that the authors are trying to do the bear minimum here to get this published, rather than develop a paper of high quality. The response letter would have been more helpful if it included the actual changes made, rather than just saying that changes had been made. The response letter made re-reviewing the paper much more time consuming than necessary. My comments are below:

Introduction - “those with a high narcissism” should read “those high in narcissim”

Method - The authors have not responded to the comment about the approving ethics committee. The question posed to them was “Which institution provided ethics approval, and what was the approval number? Please list”. The authors have just provided info on APA and Helsinki guidelines. Was this research actually approved by an ethics committee, and what is the approval number?

The following comments have not been addressed from the previous review:

- Section 2.3 – how were sports identified to participate?

- Section 2.3 – were data complete prior to, during, or after sport participation? This has implication as to whether these factors may have biased responses E.g., different patterns of response for athletes who had won / lost?

- More information is needed for the items “Satisfaction with Sporting Results”, F loser and F winner in order to properly interpret what these mean in the results. What is the reference period for these; today, past week, past year?

- “Participation was voluntary, responsible and honest.” The term responsible is not appropriate here. The researchers cannot determine is participation was “honest” without the inclusion of a lie scale. This should be deleted. This comment also points out that the term ‘responsible’ is not appropriate here – this should also have been deleted, not just “honest”.

Major criticisms:

1) The original comment to the authors read as:  The specific new contribution this study makes to the field is not entirely clear. Can the authors be specific about the new knowledge that is provided here, and why it is important? From the introduction, it seems like much of the information reported in this paper is already known to the field. I think the authors need to be clearer on what the new contribution is here. 

The response was: We consider that this question has been made clear by the explanation provided.

This reviewer disagrees – if a reader is not clear on what new information is presented then the value of publication is severely limited. I suggest the authors add some sentences to the discussion to make it crystal clear what new knowledge is being presented here.

2) The original comment to the authors read as: This may need to be re-written pending changes to the analysis.  Three of the variables were single items – this is major limitation as it is unclear how valid and reliable these measures are.

The response was: These variables seek to describe athletes' self-evaluations of their overall sports experience and what it brings to them. These are aspects that accompany the states of well-being (high satisfaction and feeling win-lost) or discomfort (low satisfaction and feeling lost). It has been decided to present them through 1 item per variable, with a scale from 0 to 10 so that a wide range of response can be offered.

There is insufficient mention of these items in the method section, and there is still no mention on these single-item variables as a limitation. Sure, they were rated on a 1-10 scale, but single item measures are often problematic from the perspective of reliability and validity – this is not discussed as a limitation.

3) This sentence in the conclusion needs to be written – the meaning of the sentence is very unclear.  “The data make the famous search for "competitiveness", apart from its obvious potential for sporting situations, a variable that promotes the appearance of dark personality traits in athletes with very competitive approaches, mainly in professional athletes.”

4) The following sentence in the conclusion is also poorly constructed and the meaning is problematic “For these reasons, according to our point of view, highlighting the type of relationship with others, the nature of the construction of narcissism, Machiavellian or psychopathic thoughts and their relationship with variables of competitiveness, would provide protective factors to athletes about their mental resources.”

The authors appear to be suggesting that traits of narcissism, Machiavellianism and psychopathic thoughts are “protective” for athletes. This is a very simplistic interpretation. And ignores the maladaptive aspects of these traits. Yes, these traits may confer some advantage in certain situations, however, they are also highly problematic from the perspective of longer-term psychological and interpersonal functioning. Those rating very high in these traits are likely to experience interpersonal problems with teammates or coaches etc. The authors need to be more sophisticated in their conclusion, and integrate what is known about the maladaptive aspects of narcissism, Machiavellianism and psychopathic traits.

Author Response

Thank you.

Reviewer 3 Report

  • The authors still have not added any specific statistics (e.g., p<0.05) to their abstract. Please add where appropriate.

Author Response

Dear reviewer 3,

We have added a specific statistics (p<0.01) to our abstract.

We have also reviewed the English language style and Spelling.

Thank you for your suggestions.

This manuscript is a resubmission of an earlier submission. The following is a list of the peer review reports and author responses from that submission.

Round 1

Reviewer 1 Report

Many thanks for the opportunity to review the manuscript 'Why negative or positive, if it make me win? Dark personality in Spanish competitive athletes' for International Journal of Environmental Research and Public Health. There is much to like about the current study and the authors should be congratulated on their efforts. Unfortunately, there are a few limitations which restrict its contribution to the literature. I have listed some concerns which should be addressed if the editor invites you to resubmit.

Abstract

Summary of the article is poor, list the exact variables and results. Aim at end of abstract is unconventional.

Introduction

lines 32-43: the rationale is unclear from this opening paragraph.

line 44: avoid generic statements (e.g., this specific construct)

Lines 62-69: not sure what this paragraph contributes

Line 107: consider revising the word pretend

Lines 107-114: aims not clear

General: the information in the intro is uninformative regarding the aims or methods.

Methods

Participants: gender of participants is missing; have the authors got any justification for their professional vs amateur split?

Instruments: missing information such as what the three subscales of the C-10 are or example items; was a translation of the scales used? is their any support or rationale for the three satisfaction or feelings scales?

Analysis: proposed analyses are inadequate - why separate ANOVA and regression when professional vs amateur could be controlled for/treated in one analysis? 

Results

Unconventional presentation of findings.

Table 2 is incomplete with values missing.

What justification are all variables entered into same MANOVA?

Terminology is inaccurate for correlations. Narcissism indeed shows the largest correlation but this does not indicate it was the most important.

Why are the other DT factors entered in the regression equation and why are all factors entered simultaneously? 

Discussion

Line 191: what previous research? 

Lines 200-237: there is very little effort to unpack the findings in these paragraphs. It is not clear how the findings related to previous research.

Line 238: conclusion is much too long and limitations should be included as its own section.

Lines 258-271: I think the conclusion should reflect the variables studied and not over reach the cross sectional design.

References

A quick view of the references point towards a review that was not focused on the main aims of the study. Additionally, recently there has been an increase in interest regarding the DT in sport, none of which are cited in the manuscript.

https://doi.org/10.1016/j.psychsport.2019.01.002

https://doi.org/10.1016/j.paid.2017.02.062

https://doi.org/10.1016/j.paid.2018.05.002

General

The structure of the article was unusual. Please adopt the style of the research above.

The writing and grasp of English wasn't high enough for publication yet. I would encourage the authors to give a thorough proof read of their manuscript before re-submission here and submission elsewhere. If the authors do not feel suitably placed to do so I would encourage them to reach out to an expert in the area of DT in sport.

I wish you all the best with publishing your valuable work in the area and it pleases me to see this type of work gaining traction. 

Reviewer 2 Report

This paper needs a full proof reading and review of English grammar, as there are errors throughout and unclear expression. This begins with the title of the article. I offer comments on the abstract and introduction. As I was not able to properly understand these sections, I ceased reading the paper at that stage. A major re-write is needed. 

Abstract

After reading the abstract, it is not clear to me what the findings of the research actually were. I highlight the below issues:

“The results show narcissism as a relevant variable, keeping same relationship with constructs on the feeling of winning as losing”. The meaning of this sentence is unclear.

 “From this perspective, and advancing the argument of the influence of dark features, we could examine the functional or dysfunctional value of dark features in athletes who have competitive characteristics compared to those who do not, in their own performance before, during and after competitions.”  I could not understand this sentence, as it suggests research that “could” be done, but the abstract should be reporting on research that has actually been done. This needs to be reworded. 

“To conclude, the aim of this paper is to understand how the personality traits…” The conclusion should not refer back to the aim of the paper, the aims needs to be stated early in the abstract. The conclusion should summarise what was discovered in the research.

Introduction

Opening sentence “Competitiveness is recognized as a behavioral achievement…” competitiveness does not necessarily relate to achievement. An individual can be competitive without actually achieving.

There is no definition offered for the key constructs of interest in this study, namely narcissism, machiavellianism and psychopathy. The reader is left unclear about what the authors mean by these terms. This is major concern.

“The study of dark triad traits (narcissism, machiavellianism and psychopathy), (DT; Paulhus & Williams, 2002) highlights the functional (Furnham, Richards, & Paulhus, 2013) and the dysfunctional (Jones, Woodman, Barlow, & Roberts, 2017) value of these traits in the athlete’s components and over those which have competitive versus non-competitive features over their own performance: before, during and after competitions.” There is no reference made to what may be functional or dysfunctional about the dark triad traits. I am not sure what the authors mean by “athlete’s components”.

“In contrast, other studies show that the competitiveness is a limitation for sports performance, which influences the appearance of a performer’s competitive behaviours (Passos et al., 2016).” I am confused what the authors are trying to communicate with this sentence.

“Most researches related to the dark side traits personality in competitive athletes which were performed with elite athletes or teams (González, Godoy-Izquierdo, & Garita-Campos, 2018), and  some of these studies suggest that athletes with a high score in competitiveness assessment, which are extremely focused on the pursuit of winning, which at the same time tend not to recognize or accept any medical advice about injuries (Zucchetti, Candela, & Villosio, 2015), generating a higher psychological (and physical) vulnerability than players which are more cautious (Chan, Gerstein, Kinsey, & Fung, 2018).” This is a very long sentence, and again, it is very difficult to understand the point being made.

“Furthermore Houston et al. (2015), established a relationship between narcissism and particularly committed to objetive behaviours, associated with disregard to the possible costs and “side effects”. Again, I do not understand what the authors are trying to communicate with this sentence.

Reviewer 3 Report

The authors have provided an interesting manuscript that provides new insights that adds to the limited research in the topic area. However there are a few aspects the authors need to consider. The information in the tables needs to be addressed. In Table 1 the numbers for General, Professional and Amateur are all the same. In Table 2 there appears to be incomplete. These will need to be amended before the manuscript can be recommended for publication. Similarly, the discussion needs integrate the findings of the present study with the previous literature.

The following comments are provided to enhance the submission further:

Tables - need to be in a standardised format

Abstract - can some more detail of the data collect and analysis methods be included?

L.13 - "in this way" is not necessary in this sentence. Consider removing

L.19 - "when athletes feel losers" may be revised to "when athletes feel like losers"

L.24-26 - rather than restating the aim can some clear conclusions from the study be provided?

L.47 - remove "they"

L.84-85 - this sentence is awkwardly written. Consider revising

L.93 - "and in facts" does not appear to fit in the sentence

L.127 - Please include a space between to and 10

L.137 - a capital letter is required for Competitiveness Scale

Procedure - please explain how the data was processed from the paper copies

Procedure - please explain how missing data was accounted for

Results - is any comparison made between and within each sport?

Discussion - greater integration of study findings are required. Currently there is also a lot of speculative information. For example L.209-214 discusses the influence of parents and coaches, however these appear to have not been assessed within the study sample. Were questions asked regarding the athletes history of parental and coaching influence?

Conclusion - this appears speculative also and fails to identify the key findings from the present study